# Dental Root Injuries Caused by Osteosynthesis Screws in Orthognathic Surgery—Comparison of Conventional Osteosynthesis and Osteosynthesis by CAD/CAM Drill Guides and Patient-Specific Implants

**DOI:** 10.3390/jpm13050706

**Published:** 2023-04-23

**Authors:** Sebastian Pietzka, Juliana Fink, Karsten Winter, Frank Wilde, Alexander Schramm, Marcel Ebeling, Robin Kasper, Andreas Sakkas

**Affiliations:** 1Department of Cranio-Maxillo-Facial-Surgery, University Hospital Ulm, 89081 Ulm, Germany; 2Department of Cranio-Maxillo-Facial-Surgery, German Armed Forces Hospital, 89081 Ulm, Germany; 3Institute of Anatomy, Medical Faculty, University of Leipzig, 04109 Leipzig, Germany

**Keywords:** orthognathic surgery, iatrogenic dental root injuries, patient-specific implant, osteosynthesis screws, virtual planning, computer assisted surgery

## Abstract

Background/Aim: The primary aim was to evaluate the prevalence and localisation of dental injuries caused by osteosynthesis screws during orthognathic surgery, comparing two different CAD/CAM planning/surgical approaches through retrospective evaluation of post-operative computed tomography. Material and Methods: This study considered all patients who underwent orthognathic surgery from 2010–2019. The examination for dental root injuries between conventional osteosynthesis (Maxilla conventional cohort) and osteosynthesis with patient-specific implant (Maxilla PSI cohort) was performed by evaluating the post-operative CT scans. Results: A total of 126 patients were included in the study. Among the 61 patients of the Maxilla conventional cohort, 10 dental root injuries in 8 patients (13.1%) were detected in the post-operative CT scan, representing 1.5% (*n* = 10/651) of the osteosynthesis screws inserted in proximity of the alveolar crest. No dental injury occurred following osteosynthesis in the 65 patients of the Maxillary PSI cohort (*n* = 0/773 screws) (*p* < 0.001). During a mean follow-up period of 13 months after primary surgery, none of the injured teeth showed evidence of periapical alterations and no endodontic treatments were necessary. Conclusions: Maxillary positioning using CAD/CAM-fabricated drill/osteotomy guide and osteosynthesis with PSI can significantly reduce the risk for dental injury compared to the conventional procedure. However, the clinical significance of the detected dental injuries was rather minor.

## 1. Introduction

Orthognathic surgery requires an interdisciplinary treatment approach between oral and maxillofacial surgeons and orthodontists [1]. The computer-assisted and virtual surgical planning with simulation and transfer of planning intraorally with CAD/CAM-manufactured surgical splints and osteosynthesis plates is increasingly gaining acceptance in clinical routine [1,2,3,4,5]. The advantages of these techniques have been demonstrated in recent years [6,7,8].

Virtual surgical planning has been improved due to the incorporation of 3D radiographic imaging and facial scans, digital dental models, digital photo-optical jaw impressions, and CAD/CAM technology. Consequently, the increased use of 3D virtual treatment planning has started to outdate conventional 2D planning in orthognathic surgery over the past decade. The benefits of this concept in terms of surgical accuracy have been recently demonstrated [1,5,8,9,10,11,12]. Of great importance is the reproducible digital 3D planning compared to model surgery in the laboratory, since the virtual imaging and planning technique enables the clinician to evaluate soft and hard tissues more appropriately [1,13,14]. Thus, individual patient anatomy can be taken into account when planning the osteotomy lines and the position and number of miniplates and, consequently, the position of the osteosynthesis screws. 

Transfer possibilities of the virtual planning to the surgical situs include CAD/CAM surgical splints [1,2,11,14,15,16], occlusally supported positioning systems [17,18], and customised osteotomy templates with PSIs [6,19,20]. Virtual 3D planning and the use of CAD/CAM-fabricated osteotomy guides, surgical splints, and PSIs have indicated accurate surgical outcomes in orthognathic surgery [1,6].

Specific complications in orthognathic surgery include, among others, the potential risk for iatrogenic dental root injury during predrilling and insertion of osteosynthesis screws [21,22,23]. Direct dental injuries can result in loss of root substance, interruption of pulpal blood supply, and root fractures. Therefore, local infections with possible fistulas, periodontal disease, tooth root resorption, radiologically osteolytic periapical or periradicular pathologies, periodontal tissue widening, and clinically hypersensitive teeth can be caused. These lesions may necessitate endodontic treatment or may also ultimately lead to tooth loss [21,22,23]. Although the surgical accuracy of 3D virtual planning with use of CAD/CAM osteotomy guides and PSIs has been often validated, no literature exists considering the safety regarding dental injury during osteosynthesis compared to conventional methods. 

The primary aim of this study was to evaluate the prevalence and location of radiologically detectable iatrogenic dental injuries after inserting osteosynthesis screws during orthognathic surgery procedures in a clinic for oral and plastic maxillofacial surgery over 10 consecutive years. The method change of maxillary positioning from conventional osteosynthesis to osteosynthesis with PSI during the study period in our clinic allows a direct comparison of possible dental injuries caused by osteosynthesis screws between the conventionally hand-bent and positioned osteosynthesis plates with the drill holes and osteotomy lines positioned via CAD/CAM templates. We hypothesised that the CAD/CAM concept would significantly reduce the risk for intraoperative dental injury. As a secondary aim, the clinical consequence of the caused dental injuries in terms of secondary need for endodontic treatment or tooth loss was also evaluated. 

## 2. Materials and Methods

### 2.1. Patient Selection

For this observational retrospective single-centre study, we reviewed the medical records of all patients who underwent bimaxillary orthognathic surgery in the clinic of oral and plastic maxillofacial surgery of the German Armed Forces Hospital Ulm between January 2010 and December 2019. Records were retrieved from our hospital electronic database. Ethical approval for this study was obtained from the ethics committee of University Ulm (approval number: 439/15). This study was performed in accordance with the Declaration of Helsinki 1964 and its later amendments (World Medical Association, Declaration of Helsinki).

We enrolled patients who fulfilled the following inclusion criteria: (1) bimaxillary orthognathic surgery; (2) pre-operative 3D virtual planning; (3) pre-operative und post-operative 3D imaging via CT; and (4) pre-and post-operative 2D orthopantomogram. Exclusion criteria were (1) no post-operative 3D imaging; and (2) incomplete medical charts. 

All surgical procedures were performed either by a board-specialised oral and maxillofacial surgeon or by a resident under supervision.

### 2.2. Surgical Planning

Computer-assisted, virtual 3D surgical planning was performed using ProPlan CMF^®^ planning software from Materialise© (Leuven, Belgium). A high-resolution CT with reconstruction in 1-mm slices was required. In the maxilla, the Le Fort I osteotomy line was digitally designed and, in the mandible, according to Obwegeser-Dal Pont sagittal split method. After the digital osteotomy, the individual segments could be freely moved in all three spatial planes during computer planning. In addition, the displacement distances could be determined.

The difference between the two surgical methods for maxillary positioning is the different transfer of the 3D planning into the surgical site. Conventionally hand-bent and manually positioned osteosynthesis plates were used in the study period from 2010–2014 and CAD/CAM-manufactured drilling/osteotomy templates with patient-specific implants for osteosynthesis were used in the study period 2015–2019. 

#### 2.2.1. Maxillary Positioning Using CAD/CAM Intermediate Splint, Temporary Mandibular Fixation and Conventional Hand-Bent and Manually Positioned Osteosynthesis Plates

After 3D planning has been completed, plaster models of the maxilla and mandible are scanned and matched with the CT dataset by a medical engineer and act as the basis for the creation of the CAD/CAM-manufactured intermediate splint. The intermediate splint is used for intraoperative fixation of the new maxillary position in relation to the mandible. To avoid mandible mobility, a temporary mandibular fixation by means of fixation plates is aimed at for positioning the maxilla, which corresponds to the mandibular position in the initial planning CT. For this purpose, an initial splint is fabricated to reproduce the pre-operative position of the mandible and is in situ during the pre-operative initial CT and intraoperatively at the start of the adjustment osteotomy. Intraoperatively, the mandibular position defined in this way is secured to the zygomatic region with fixation plates from the mandibular ramus. These can be removed for the maxillary osteotomy and then reproducibly fixed again via the drill holes. The intermediate splint, designed by Materialise© (Leuven, Belgium), is computer-assisted and subsequently 3D-printed. The initial and target splints, which encode the target occlusion, are conventionally fabricated in the articulator in the dental laboratory (Pietzka). 

After Le Fort I osteotomy, the ascending mandibular branch and the mandibular angle are exposed and mandibulomaxillary fixation with wire ligatures is followed. An L-shaped titanium plate with four osteosynthesis screws is placed bilaterally to fix the ascending mandibular branch and the crista zygomaticoalveolaris cranial to the pre-osteotomised maxilla in order to complete the osteotomy. After removal of the fixation plates, the maxilla can be freely mobilised and adjusted with the CAD/CAM intermediate splint, following mandibulomaxillary fixation over the orthodontic arches. The previously removed L-shaped fixation plates are reinserted bilaterally in the original position using the former drill holes, and the planned maxillary position is transferred to the surgical site via the CAD/CAM splint. The conventional osteosynthesis starts with a hand-bent L-shaped plate on each side paranasally. The fixation plates are then removed again and the definitive, conventional osteosynthesis in the region of the crista zygomaticoalveolaris is also performed with a hand-bent L-shaped plate (DePuy Synthes^®^, Matrix ORTHOGNATHIC^TM^, Warsaw, IN, USA) and monocortical osteosynthesis screws with a diameter of 1.85 mm and a typical length of 4 mm bilaterally (Figure 1).

#### 2.2.2. Maxillary Positioning Using CAD/CAM-fabricated Drill/Osteotomy Guide and Osteosynthesis with PSI

The transfer of the planned maxillary position into the surgical site is conducted waferless with a laser-melted CAD/CAM drilling/osteotomy template and a laser-melted CAD/CAM PSI. 

The drilling/osteotomy template and the PSI was designed and manufactured by Materialise© (Leuven, Belgium) after an online planning session with the responsible surgeon team. In the planning, the pre-drilled holes and the osteosynthesis screws were positioned in such a way that the tooth roots were spared (Figure 2). The drilling/osteotomy template is passively and temporarily fixed to the maxillary bone with two to a maximum of four additional osteosynthesis screws. The holes for the subsequent osteosynthesis are then drilled both cranially and caudally to the osteotomy using the drill/osteotomy templates preformed drill channels. The Le Fort I osteotomy line is guided in the area of the anterior and lateral walls of the maxillary sinus as virtually planned and is completed after removal of the drill/osteotomy template. After “down fracture”, the patient-specific implant is passively placed on the caudal mobile maxilla and fixed with screws through the already predrilled holes. The design and shape of the patient-specific implant encodes the final position of the maxilla in all three planes with a secure and reproducible fit (Figure 3). Therefore, the maxillary positioning can be performed intraoperatively, independently of the mobile mandible. The maxilla is moved cranially until the empty screw holes of the PSI are also passively positioned over the pre-drilled holes in the cranial part on both sides of the midface. The PSI fixation follows by using monocortical osteosynthesis screws with a diameter of 1.85 mm and a typical length of 4 mm (Figure 4). 

### 2.3. Mandibular Realignment

Osteotomy procedure in the mandible was the same for both surgical methods in the maxilla. After sagittal splitting according to Obwegeser–Dal Pont and sufficient mobilisation of the three mandibular segments, the target splint was integrated between the maxilla and mandible using mandibulomaxillary fixation via wire ligatures over the orthodontic arches. Osteosynthesis was now regularly performed using SplitFix plates (DePuy Synthes^®^, Matrix ORTHOGNATHIC^TM^, USA). Monocortical osteosynthesis screws (DePuy Synthes^®^, Compact 2.0, USA) with a diameter of 2 mm and a length of 6 mm were typically used (Figure 5).

### 2.4. Data Collection

Due to the CAD/CAM planning of all bimaxillary orthognathic cases, all patients of the observation period were known and already listed. Further data acquisition was carried out separately for each individual patient cases by one single doctor. This doctor evaluated the available pre- and post-operative radiographic findings and examined them for possible dental injuries. All CT datasets with a close tooth–screw relationship were then viewed together with a specialist in maxillofacial surgery using the four-eye principle. The other data were taken from the patients’ medical records and the hospital information system, and the patients were anonymised before data analysis.

Patients enrolled were subsequently divided into two groups according to the surgical method used for maxillary positioning: (1) maxillary positioning via CAD/CAM intermediate splint, temporary mandibular fixation, and conventional osteosynthesis (Maxilla Conventional) in 61 patients; and (2) splint-free maxillary positioning via CAD/CAM drilling/osteotomy template and osteosynthesis using patient-specific implant (Maxilla PSI) in 65 patients. 

The maxilla was evaluated separately from the mandible. 

In the mandible, no grouping was actually necessary, as the surgical procedure in the mandible did not differ from the two surgical methods in the maxilla. Nevertheless, the mandible was additionally considered separately in both groups. In this way, it should be possible to identify surgeon-dependent bias in the results for the upper jaw as well. Indication for orthographic surgery and the individual treatment plans were not evaluated as they were irrelevant to the aim of the study. Only the osteosynthesis screws inserted proximately to the tooth, i.e., caudal to the Le Fort I osteotomy, were included in the investigation.

### 2.5. Radiological Evaluation

The performed pre-operative and post-operative radiographic imaging was viewed in the viewing program Visage^®^ 7 Viewer (Visage Imaging^®^, San Diego, CA, USA) on an X-ray reporting monitor. The findings obtained were documented in a recording sheet. 

Dental injury was defined as the presence of an osteosynthesis screw clearly within a dental root; the mere touching of the root surface detected on the CT scan was not considered as dental injury. In addition, classification of root injuries with or without pulp involvement was performed. In cases with possible contact between the screw and the dental root in the post-operative CT imaging, the scan was additionally and independently examined by a second examiner of the same clinic. 

In each patient, the pre-operative orthopantomogram was assessed in terms of periapical alterations and previous endodontic treatments. In the post-operative orthopantomogram, the number and position of the inserted osteosynthesis screws were recorded in addition to the surgical report. The radiological projections of the inserted screws to the dental roots were also examined. When osteosynthesis screws were projected on a dental root, it was recorded as a possible dental injury in the associated tooth region. Radiologically detected root lesions were documented according to the tooth region (anterior/canine/premolar/molar) and the root localisation. In the maxilla, only the osteosynthesis screws placed caudally to the osteotomy line were evaluated for possible dental injury. The results of post-operative panoramic images and CT scans were compared. In cases of radiological projection of osteosynthesis screw on dental roots on post-operative panoramic radiographs, verification was made by examining the post-operative CT scans. The post-operative CT scans were evaluated in the axial, coronal, and sagittal planes.

The following areas were distinguished in the maxilla: anterior region (teeth 12–22), canine region (teeth 13 and 23), premolar region (teeth 14, 15 and 24, 25), molar region (teeth 16, 17, 18 and 26, 27, 28), and the midface region cranial to the osteotomy line. In the mandible, the distinction is made between the premolar region (teeth 34,35 and 44,45), the molar region (teeth 36, 37, 38 and 46, 47, 48), and outside the dentition. Cases of genioplasty were also documented regarding screw projections and dental injuries in the mandibular anterior region.

Patients with dental injuries were evaluated post-operatively as part of the normal aftercare and not additionally to the regular intervals. In all cases with CT-diagnosed dental root injuries, the orthopantomograms were additionally evaluated before and after removal of the osteosynthesis material, usually 6 months to 1 year after primary surgery. Focus was given to alterations in the apical region. Further evaluation criteria were assessment of the periodontal gap, bone resorption, and tooth loss. The panoramic radiographs before osteosynthesis material removal were compared simultaneously onto a radiographic evaluation monitor with the pre-operative scans. After removal of the osteosynthesis material, no long-term follow-up examination of the injured teeth was performed in our clinic.

### 2.6. Statistical Analysis

Data were centralised in electronic format using Microsoft Excel 2019 software (Microsoft Corporation, Redmond, WA, USA) and analysed descriptively. Statistical analysis was performed using IBM SPSS^®^ Statistics 26.0 (IBM, Armonk, NY, USA). Metric data were expressed as mean and standard deviation (SD), while nominal data were expressed as frequency and percentage. Descriptive statistics were used to describe baseline patient characteristics. All categorical variables were expressed as absolute values (*n*) and relative prevalences (%). Absolute frequencies were calculated using the Fisher’s Exact Test. The significance level was set at *p* < 0.05. The odds ratio was used to calculate the risk for dental injury in the maxilla in the Maxilla Conventional and Maxilla PSI cohort. 

## 3. Results

### 3.1. Patient Collective

During the observation period, 129 patients underwent 3D planned bimaxillary orthognathic surgery at the German Armed Forces Hospital in Ulm. This retrospective study included 126 patients who met the inclusion criteria. Three patients could not be included due to lack of post-operative 3D imaging. There were more males (*n* = 65; 51.5%) than females (*n* = 61; 48.5%) (male to female ratio = 1.06:1). Patient age at the time of surgery ranged from 17.7 to 60.7 years, with a mean age of 24.9 years.

### 3.2. Analysis of the Inserted Osteosynthesis Screws

A total of 2734 osteosynthesis screws inserted into the maxilla were documented. Among them, 1310 screws were inserted cranially to the Le Fort I osteotomy line and 1424 caudally. This screws below of the osteotomy line are consequently close to the alveolar ridge and the dental roots. In the Maxilla Conventional cohort, 1191 osteosynthesis screws were inserted with a mean of 19.5 screws (median 18, min. 16, max. 28) in 4 4- or 6-hole osteosynthesis plates per patient. In the Maxilla PSI cohort, 1543 osteosynthesis screws were inserted with a mean of 23.7 screws (median 24, min. 21, max. 26) per patient case. 

In the mandible, a total of 1037 osteosynthesis screws were inserted in both cohorts, with a mean of 8.2 screws (median 8, min. 8, max. 23) per patient case. Among them, 454 screws were inserted outside the tooth row (distally to the last molars), 37 in the premolar region, and 546 in the molar region.

### 3.3. CT Diagnosed Dental Injuries 

Ten dental root injuries in eight patients were detected in the post-operative CT scan in the Maxilla Conventional cohort (*n* = 8/61 patients; 13.1%). This represents 1.5% of the osteosynthesis screws inserted proximately to the alveolar crest (*n* = 10/651). Two patients sustained a double injury. No dental injury occurred in 65 patients of the Maxillary PSI cohort (*n* = 0/773 osteosynthesis screws). A higher risk for iatrogenic root injuries was shown in patients with conventional osteosynthesis of the maxilla compared to osteosynthesis with PSI (Mc Nemar test: *p*< 0.001; Odds ratio = 0). Regarding the mandible, 12 dental root injuries (*n* = 12/583; 2.05%) in 10 patients were documented in the post-operative CT scan (*n* = 10/65 patients; 15.3%). Among them, three were detected in premolars and nine in molars. No significant difference was observed regarding mandibular dental injuries between the two operation method cohorts (Mc Nemar test: *p* = 0.195). 

Table 1 shows the distribution of dental root injuries in both maxilla and the mandible in regard to the operation method for maxillary positioning and the dental region. Table 2 shows the dental injuries in both maxilla and the mandible in regard to the root location and pulp involvement. 

### 3.4. Comparison of Projection of Osteosynthesis Screws with Dental Roots in Post-Operative OPG and CT Scans

Figure 6 and Figure 7 present a patient case after conventional maxillary positioning, showing projections of osteosynthesis screws with the dental roots in the OPG and, subsequently, the confirmed or not confirmed dental injuries in the CT scan.

In the Maxilla Conventional cohort, 94 radiological projections of osteosynthesis screws with the dental roots were detected in the post-operative OPG. In contrast, post-operative CT imaging detected only 10 dental root injuries. In the Maxilla PSI cohort, 64 radiological projections were detected; however, no dental root injury could be detected in post-operative CT imaging. In the mandible, a total of 106 projections in the premolar and molar region were recorded in the post-operative OPG, but only 12 root lesions were detected in post-operative CT scan. 

Table 3 presents the total number of projections of osteosynthesis screws with dental roots in the post-operative OPG and the number of confirmed root injuries via CT in the maxilla regarding the dental region and the surgical method used. 

### 3.5. Follow-up of the Dental Injuries

A follow-up of patients with confirmed dental root injuries was conducted at 13 months (median 9.9, min. 6.1, max. 45.5). During this period, none of the injured teeth showed evidence of periapical alterations and no endodontic treatments were performed. Figure 8 presents a case with CT-confirmed dental root injury directly, post-operatively, and after removal of the osteosynthesis material 10.8 months after primary surgery.

## 4. Discussion

We conducted this study in an oral and maxillofacial surgery clinic to evaluate the influence of two different osteotomy and osteosynthesis surgical methods for maxillary positioning on the incidence of iatrogenic dental root injuries caused by osteosynthesis screws during bimaxillary orthognathic surgery. The hypothesis was that maxillary positioning using CAD/CAM drill/osteotomy guide and osteosynthesis with PSI would significantly reduce the risk of dental injury compared to conventional osteosynthesis. This research represents a considerable collective and reflects the need to specify one more possible advantage of CAD/CAM-guided maxillary osteotomy and osteosynthesis with PSI over the conventional method. 

Dental root injury after insertion of osteosynthesis screws in both fracture treatment and in orthognathic surgery is a commonly known complication. While tooth injuries after mandibulomaxillary fixation during fracture treatment by IMF-screws have been adequately investigated in the past, only a few studies could be identified that investigated dental injuries caused by osteosynthesis screws after trauma or orthognathic surgery [23,24,25,26,27,28]. To our knowledge, this is the first study to evaluate dental injuries solely during orthognathic surgery, namely, by comparing them to different maxillary positioning surgical methods. In the maxillary conventional group, definitive osteosynthesis was performed using four hand-bent L-plates and unguided screw positioning. In the maxillary PSI group, the planning was transferred to the surgical site using a CAD/CAM drilling/osteotomy template and guided osteosynthesis with a patient-specific implant.

Comparative literature reported incidences of tooth root injury from inserted osteosynthesis screws ranging from 0.28% to 2.3%. In the study by Driemel et al., dental injury occurred after insertion of 30 screws out of 2100 screws [21]. Balaji et al. documented five dental injuries after insertion of 4472 screws during orthognathic surgery [29]. Pabst et al. detected 16 (0.5%) dental injuries in 13 out of 366 patients in a mixed collective including trauma treatment and orthognathic surgery [30]. He also postulated a distance of 10 mm from the nearest dental root as considerably safe, avoiding root damage. We detected a low incidence of 0.7% of dental injuries caused by osteosynthesis screws in the maxilla. Our results are in concordance with the above-mentioned literature; however, a direct comparison is limited due to the different collective and study design. Considering the mandible, osteosynthesis was performed conventionally with SplitFix plates after conversion with the already fixed maxilla through the final splint. Thus, we think that a separate comparison of dental injuries between the two maxillary surgical methods is not reasonable. 

A possible limitation of the comparability of our two maxillary cohorts might be a difference in the experience of the surgeons. Even though the surgeons in both groups were assigned in the same way, the individual procedures were performed by experienced specialists, but also by residents under specialist supervision. Therefore, we considered the injury incidence in the two mandibular cohorts in order to identify a possible surgeon-related bias and detected a relatively low incidence (2.06%) of mandibular dental injury. There were dental injuries in both mandibular cohorts. As expected, no significant difference was detected between the two cohorts in mandibular treatment. Therefore, the significant differences between the two cohorts in the maxilla will certainly not be surgeon-dependent but determined by the methods. 

The separate evaluation of the two surgical methods showed a clear advantage of maxillary osteosynthesis with PSI compared to the conventional procedure, since a dental injury could be prevented in all cases. We demonstrated maxillary positioning with CAD/CAM drill/osteotomy guide and osteosynthesis with PSI as an absolutely safe surgical method for all surgeons, independent to their experience level. 

This method was initially introduced for waferless maxillary positioning intending to increase accuracy and predictability, as demonstrated by Heufelder et al. [31]. While these advantages have already been sufficiently demonstrated and validated, the prevention of tooth damage appears now as an additional benefit for the surgical outcome. This raises the interest in development of a similar guided procedure for mandibular osteotomy and osteosynthesis as potentially beneficial regarding root and nerve damage prevention. 

No standard classification of dental root injuries caused by inserted osteosynthesis screws exists in the literature. Driemel et al. and Pabst et al. described in their studies different injury patterns considering the involvement of pulp; however, this was not the aim of our study [21,30]. We documented root injuries after clear radiological injury confirmation via CT imaging. The evaluation of the orthopantomograms in this study was performed only to obtain an overview of the number of screws and possible tooth damage due to close positional correlation. Previous studies also evaluated tooth root injuries based on 2D radiographic imaging [21,29,32]. Examination using 2D imaging was found to be inaccurate compared to 3D imaging in the study of Kauke et al. [33]. He reported that a conventional dental film or orthopantomograms cannot reveal the true depth of penetration of the inserted screws or the involvement of the dental pulp, leading to false positive results [21,24,25,34]. Our results confirm that the extension of the bony defect, or root lesions per se, can be accurately visualised only with 3D radiographic imaging, such as via CT or CBCT [23,30,35]. In principle, intraoperative imaging via a 3D C-arm in orthognathic surgery would also be conceivable here, particularly since these are already regularly used in traumatology and our own studies show evidence of a reduced effective radiation dose compared to post-operative 3D X-ray procedures. Pabst et al. could verify using CTBC only 16 out of 230 dental root lesions assumed on OPG [30]. This correlates with the data of the present work. In both the maxilla and mandible, the OPG evaluation showed significantly more root lesions than were verified via CT. No additional injuries were detected with CT scans compared to OPGs. This was to be expected, as the evaluation of OPGs as an examination tool for the detection of dental injuries is rather limited due to the extremely high number of false positive indications of dental damage and the correspondingly low specificity. We suggest that orthopantomograms alone are not suitable for detecting dental injuries caused by inserted osteosynthesis screws, and 3D imaging can proof diagnosis in necessary cases. However, its indication to assess the position of the osteosynthesis material, the osteotomy course, and the condylar position post-operatively remains valid. 

Regarding the location of tooth injury in our collective, no “safe zone” could be identified. The distribution of tooth root injuries was approximately equal in all four quadrants and most likely randomly distributed. Only the study of Balaji et al. differentiates the location of the tooth root injury per tooth region after orthognathic surgery similar to the present study, albeit not between the maxilla and mandible [29]. He detected injuries in three molars and two premolars, while we documented eighteen injured molars, three premolars and one canine. The distribution of injuries between molars and premolars was quite even; however, a valid comparison of data is only possible to a very limited extent due to the small number of tooth injuries. Furthermore, the higher number of injuries in the molar region is explained by the significantly lower number of inserted osteosynthesis screws in the premolar and anterior regions. Contrary to what was assumed, canine injuries occurred only once in our study, especially in the upper jaw - the canine is the tooth with the longest root.. Al-Jandan et al. stated in his investigation that the mandibular buccal bone thickness increases posteriorly and inferiorly, except for the second premolar [36]. This is consistent with the present results, which show an increased risk of injury in this region after the detection of three premolars with root lesions on CT among six premolars with projection on OPG. Considering the findings of Al-Jandan et al., we did not record root injuries of second molars. This could also be explained because the vertical osteotomy line is placed between the first and second molars, or at the level of the second molar, and thus there is per se less, if any, overlapping of osteosynthesis screws with the second molars. 

In the present study, none of the teeth with CT-confirmed injury developed apical alterations or root fracture and no endodontic treatment or extractions were performed during the observation period. No consistent clinical data exist regarding the development of apical pathologies after dental trauma with osteosynthesis screws. Several animal studies have demonstrated that peripheral dental root lesions, without direct pulp involvement, can often heal without sequelae, and can partially heal with complete repair of periodontal structures [37,38,39]. On the contrary, partial or no healing was seen in teeth with pulp involvement [37]. Abbott et al. indicates that trauma-related pulpal necrosis and infection usually occurs within the first three to four months; however, this was related to orthognathic or trauma surgery specifically [40]. According to Driemel et al., only root injuries involving the dental pulp increase the risk of pulpitis and can consequently result in apical periodontitis and tooth loss [21]. In contrast, injuries without pulp involvement have a good prognosis. In his study, he detected radiographic alterations such as periapical translucencies, replacement resorptions, or external resorptions in all 19 root injuries. These changes occurred in a mean period of 18 months and eight of the teeth with pulp involvement required endodontic treatment. Our results are not in accordance with Driemel et al., since none of the 22 injured teeth, with or without pulp involvement, developed radiologically detectable periapical pathologies. However, our follow-up period was significantly shorter. Similarly, Borah et al. also did not record any endodontic treatment in 11 out of 13 patients with dental injuries in a follow-up of 33 months [32]. In the study by Pabst et al., four patients with dental injury were clinically observed in a non-systematic follow-up [30]. Three of these teeth were devitalised, one was extracted, and two underwent endodontic treatment. Since our study design did not aim a long-term follow-up over the observation period, we cannot rule out late development of pulpal necrosis or apical alterations with need for endodontic treatment. Symptoms of iatrogenically damaged teeth can occur even after decades, especially in young patients [40]. Studies of dental injuries following insertion of IMF screws indicated that the extension of the follow-up period has no impact on the need for endodontic treatment [23,27,33,41]. However, we suggest clinical evaluation of the affected teeth in every regular consultation until completion of the surgical treatment. It has also to be considered that sensitivity tests can only be of limited significance regarding endodontic treatment, especially in cases with neural disorders in the spread area of the inferior alveolar or the infraorbital nerve after osteotomy [31]. 

Although we observed a rather low clinical relevance of iatrogenic tooth root injury caused by inserted osteosynthesis screws in this collective, we recommend continuous clinical evaluation through the family dentist after completion of the surgical treatment.

Previous studies showed a high predictability and accuracy in maxillary positioning using CAD/CAM-fabricated individual drilling/osteotomy templates and patient-specific osteosynthesis plates, as well as a reduction of the surgical time compared to conventional methods [31]. The additional benefit of the clear significant risk reduction for iatrogenic dental root injuries, at least during maxillary positioning, must be contrasted by the additional costs of approx. 2.000–3.000 euros compared to the simple commercially available conventional osteosynthesis miniplates. The exact cost comparison for a valid risk/benefit evaluation was not the aim of this study. To date, the high costs of 3D planned orthognathic surgery with CAD/CAM components are not covered or have only been partially covered by health insurance companies. Consequently, the routine use of this method is limited in most of the clinics. However, our work highlights the high degree of planning reliability and implementation in the surgical site for the avoidance of dental injuries, and encourages the application of this method, especially in complex patient cases. We further recommend this surgical method as significant in training since the visualised planning process and the highly reproducible results can introduce junior surgeons to orthognathic surgery. 

There are some limitations to the current study. First, the retrospective nature of the research could lead to documentation bias at the defined time points after osteotomy and before material removal. However, this is outweighed by the sufficient patient number. The absence of studies with similar design did not allow a direct comparison regarding the incidence of tooth root injuries due to the osteosynthesis screws following two different planning/surgical methods, which limits the generalisability of our results. Another limitation of this work is that patients with dental injuries were only radiologically observed directly post-operatively, during the regular post-operative consultations, and before the removal of osteosynthesis material. Separate routine clinical examination, including pulp vitality testing, percussion test, and probing depth, was not performed at regular intervals in our clinic during the follow-up period and after material removal. During the development of the study design, a conscious decision was made not to perform additional clinical examinations due to the wide catchment area of patients and the different primary surgical time points within the 10-year study period. We assume that further clinical examinations might have detected cases with need of endodontic treatment. A classification of different root injury patterns in terms of location and extent was also not performed in this study, and it could be helpful to associate the extension of injury with the long-term prognosis. Future well-designed studies with the prospective protocol are required to validate our preliminary results for clinical practise.

## 5. Conclusions

We conclude that maxillary positioning using the CAD/CAM drill/osteotomy guides and osteosynthesis with PSI can significantly reduce the risk of dental injury compared to the conventional method with hand-bent and manually positioned osteosynthesis plates. We highlight the additional benefit of this method in orthognathic surgery; however, cost effectiveness has to be critically considered. The clinical significance of the detected dental injuries in our collective was rather minor. We suggest that orthopantomograms alone are not suitable for the reliable detection of dental injuries, and only 3D imaging can assist diagnosis in necessary cases. Radiologically confirmed dental injuries should be long-term monitored in order to initiate endodontic treatment at an early stage if necessary.

## Figures and Tables

**Figure 1 jpm-13-00706-f001:**
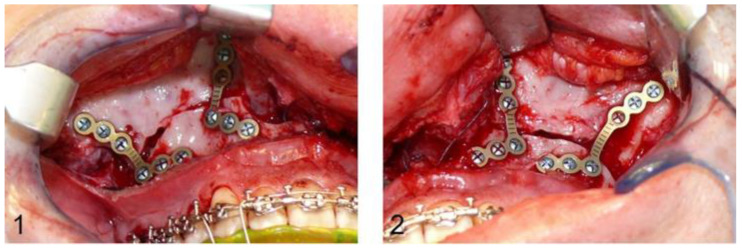
Image (**1**,**2**): Intraoperative situs after definitive conventional osteosynthesis of the maxilla with L-shaped osteosynthesis plates paranasally and in the region of the crista zygomaticoalveolaris.

**Figure 2 jpm-13-00706-f002:**
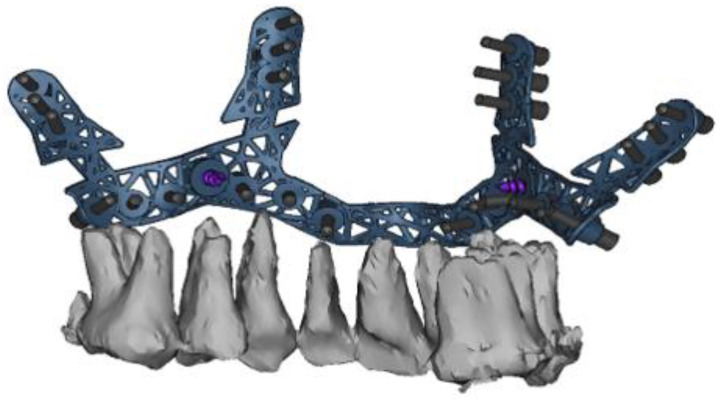
Virtual computer-assisted planned drill/osteotomy guide with protection and consideration of the maxillary dental roots.

**Figure 3 jpm-13-00706-f003:**
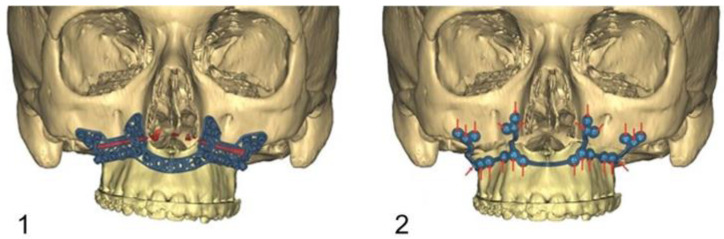
Image (**1**): Computer assisted designed drill/osteotomy template with the Le Fort I osteotomy line (red), the drill channels for the temporary fixation of the template, and the drill channels for the subsequent definitive osteosynthesis with PSI. Image (**2**): New positioned maxilla after definitive osteosynthesis with PSI (red arrows= inserted osteosynthesis screws).

**Figure 4 jpm-13-00706-f004:**
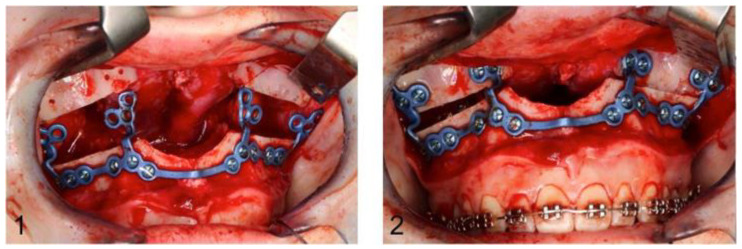
Image (**1**): PSI fixed with osteosynthesis screws caudally to the maxillary osteotomy line. Image (**2**): Definitive fixation of the PSI after cranial mobilisation of the maxilla until it overlaps the predrilled holes.

**Figure 5 jpm-13-00706-f005:**
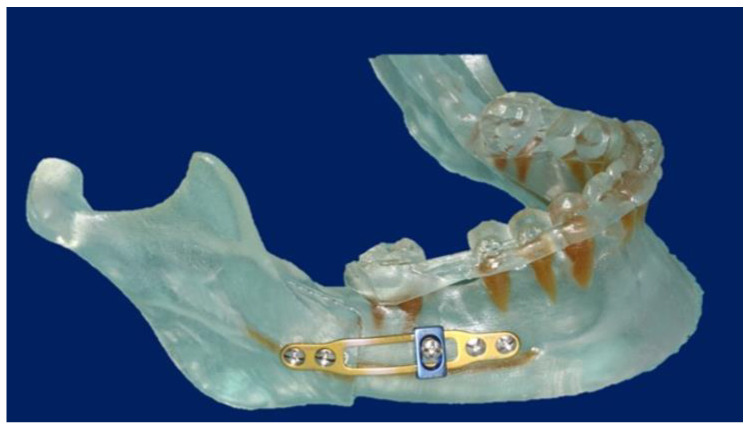
Definitive osteosynthesis with two screws: mesially and distally to the osteotomy line, respectively.

**Figure 6 jpm-13-00706-f006:**
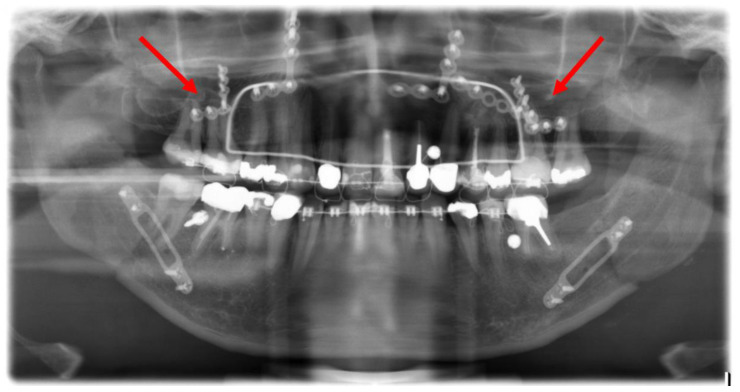
Post-operative orthopantomogram after the conventional maxillary positioning, showing the projections of three osteosynthesis screws with the dental roots of the teeth 16 and 17 and the projections of four osteosynthesis screws with the dental roots of the teeth 26 and 27 (red arrows).

**Figure 7 jpm-13-00706-f007:**
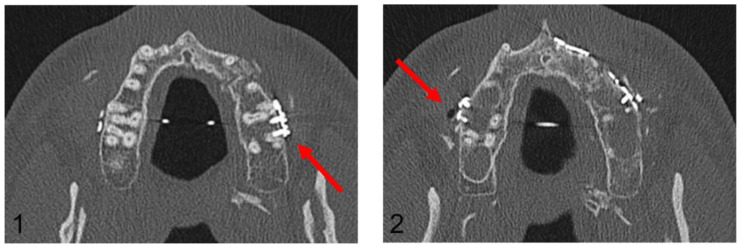
Axial plane of post-operative CT scan. Image (**1**) shows an injury of the mesiobuccal tooth root of tooth 27, while in image (**2**), no root injury in the right maxillary molars is detected (red arrows).

**Figure 8 jpm-13-00706-f008:**
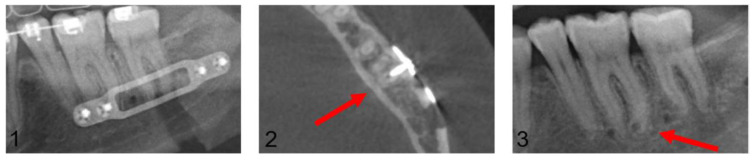
Image (**1**): Projection of an osteosynthesis screw in the distal root of tooth 36 in the post-operative orthopantomogram after mandibular sagittal osteotomy. Image (**2**): CT confirmed dental root injury of the distal root of tooth 36 (red arrow). Image (**3**): Left posterior mandibular region after removal of the osteosynthesis material 10.8 months after primary surgery. The red arrow marks the area of radiologically proven root injury. No apical alterations are shown, and no endodontic treatment had been necessary up to this point.

**Table 1 jpm-13-00706-t001:** Distribution of dental root injuries in both maxilla and the mandible in regard to the operation method for maxillary positioning (Maxilla Conventional / Maxilla PSI) and the dental region.

	Maxilla Conventional		Maxilla PSI	
**Maxilla**	**Root Injuries/Total (N)**	**%**	**Root Injuries/Total (N)**	**%**
anterior	0/147	0.00%	0/203	0%
canine region	1/109	0.92%	0/118	0%
premolar region	0/180	0.00%	0/186	0%
molar region	9/215	4.19%	0/266	0%
Total	10/651 *	1.54%	0/773	0% *
**Mandible**				
premolar region	2/20	10%	1/17	5.88%
molar region	6/261	2.29%	3/285	1.05%
Total	8/281 **	2.84%	4/302 **	1.32%

Abbreviations: N = number; * Mc Nemar test: *p* = *p* < 0.001; ** Mc Nemar test: *p*= 0.195.

**Table 2 jpm-13-00706-t002:** Dental injuries in the maxilla and mandible with respect to the root location and pulp involvement.

Maxilla	N	Mandibula	N
13	1	35	1
16 distobuccal	2	45	2
16 mesiobuccal	1	36 distal	3
17 distobuccal	0	36 mesial	3
17 mesiobuccal	2	46 distal	1
26 distobuccal	2	46 mesial	2
26 mesiobuccal	1		
27 distobuccal	0		
27 mesiobuccal	1		
Total	10	Total	12
pulp involvement	5	pulp involvement	4
no pulp involvement	5	no pulp involvement	8

Abbreviations. N = number.

**Table 3 jpm-13-00706-t003:** Distribution of projections of osteosynthesis screws with dental roots in the post-operative OPG and the confirmed root injuries via CT in both maxilla and the mandible regarding the dental region and the surgical method used.

Tooth Region	Projections (N) OPG	Dental Injuries (N) CT
	**Maxilla Conventional**	
**anterior**	3	0
**canine**	11	1
**premolar**	10	0
**molar**	70	9
**Total**	94	10
	**Maxilla PSI**	
**anterior**	3	0
**canine**	3	0
**premolar**	7	0
**molar**	51	0
**Total**	64	0
	**Mandible**	
**premolar**	6	3
**molar**	100	9
**Total**	106	12

Abbreviations: N = number; OPG = orthopantomogram; CT = computer tomography.

## Data Availability

The datasets generated and analysed during the current study are not publicly available due to institutional restrictions but are available from the corresponding author upon reasonable request.

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
