# Peer review of "Dental Root Injuries Caused by Osteosynthesis Screws in Orthognathic Surgery—Comparison of Conventional Osteosynthesis and Osteosynthesis by CAD/CAM Drill Guides and Patient-Specific Implants"

_jpm, 2023, doi:10.3390/jpm13050706_

Round 1

Reviewer 1 Report

Thsmk you for the invitation  to review this paper.

Generally, it is as well written article with an important topic.

There are few monor issues:

The presentation of the OPG to CT accuracy is complicated and not well understood. I don't think that these calculations have any significance because of the small number of injuries overall. Maybe in should be dropped from the study at all.

There is a limitation that you didn't mention - the fact that the sugeries were performed by different surgeons and some parts of them by trainees. It is a major weakness of the study and should clearly mentioned.

Minor corrections - line 66 - add commas before asnd after "among other".

Line 401 - "diffeternt" should be changed to "difference"

Author Response

Dear respected colleague,

thank you very much for reviewing our article. We would also like to thank you  for your constructive and useful comments.

We have adapted our manuscript according to your recommendations. You can follow the changes in word.

We have removed the calculation of senisivity and specificity of OPG vs CT.

We also clarified the limitation that both specialists and residents performed the operations in both groups under supervision.

However, we see this as a particular advantage of the PSI method, as no tooth damage occurred here, although trainees also performed the operations in this group. When operating on the lower jaw without PSI, however, tooth damage occurred in both cohorts of the upper jaw. Thus, the highly significant differences in the upper jaw seem to be independent of the experience of the surgeons and are only due to the surgical method.

Thank you very much.

Kind regards,

Sebastian Pietzka

Reviewer 2 Report

Dear Authors,

Thanks for this good job, I found the topic interesting. Following you can find some feedback:

1. Include more Oral and maxillofacial surgery services could develop a multicenter study. This new study would have more reliable results. Perhaps you can recommend it for future studies and include this aspect within the weaknesses of the study.

2 at Methods section required more description in "data collection"  (only)

Author Response

Dear respected colleague,

thank you very much for reviewing our article. We would also like to thank you  for your constructive and useful comments.

We have adapted our manuscript according to your recommendations. You can follow the changes in the word document.

In particular, we have now explained the data collection in more detail.

We will take up your idea of a multicentre study on this topic. Unfortunately, splintless maxillary repositioning with patient-specific implant and drill guide is still rather rare in Germany. Hand-bent osteosynthesis plates (and presumably the associated tooth damage) are still common in the majority. We hope that our work will motivate other colleagues to use the new procedures.

Kind regards 

Sebastian Pietzka